# Malaria Temporal Variation and Modelling Using Time-Series in Sussundenga District, Mozambique

**DOI:** 10.3390/ijerph18115692

**Published:** 2021-05-26

**Authors:** João L. Ferrão, Dominique Earland, Anísio Novela, Roberto Mendes, Alberto Tungadza, Kelly M. Searle

**Affiliations:** 1Instiuto Superior de Ciências de Educação, Beira 2102, Mozambique; 2School of Public Health, University of Minnesota, Minneapolis, MN 55455, USA; earla001@umn.edu (D.E.); ksearle@umn.edu (K.M.S.); 3Direcção Distrital de Saúde de Sussundenga, Sussundenga 2207, Mozambique; novela.anisio@hotmail.com; 4Centro de Informação Geográfica-Faculdade de Economia da UCM, Beira 2102, Mozambique; rmendes@ucm.ac.mz; 5Faculdade de Ciência de Saúde da UCM, Beira 2102, Mozambique; atungadza@gmail.com

**Keywords:** malaria, modelling, temporal, Sussundenga

## Abstract

Malaria is one of the leading causes of morbidity and mortality in Mozambique, which has the fifth highest prevalence in the world. Sussundenga District in Manica Province has documented high *P. falciparum* incidence at the local rural health center (RHC). This study’s objective was to analyze the *P. falciparum* temporal variation and model its pattern in Sussundenga District, Mozambique. Data from weekly epidemiological bulletins (BES) was collected from 2015 to 2019 and a time-series analysis was applied. For temporal modeling, a Box-Jenkins method was used with an autoregressive integrated moving average (ARIMA). Over the study period, 372,498 cases of *P. falciparum* were recorded in Sussundenga. There were weekly and yearly variations in incidence overall (*p* < 0.001). Children under five years had decreased malaria tendency, while patients over five years had an increased tendency. The ARIMA (2,2,1) (1,1,1) _52_ model presented the least Root Mean Square being the most appropriate for forecasting. The goodness of fit was 68.15% for malaria patients less than five years old and 73.2% for malaria patients over five years old. The findings indicate that cases are decreasing among individuals less than five years and are increasing slightly in those older than five years. The *P. falciparum* case occurrence has a weekly temporal pattern peaking during the wet season. Based on the spatial and temporal distribution using ARIMA modelling, more efficient strategies that target this seasonality can be implemented to reduce the overall malaria burden in both Sussundenga District and regionally.

## 1. Background

Malaria is an ancient disease that occupies a unique place in the annals of history. Globally there were 228 million malaria cases recorded in 2018. The Sub-Saharan African region has the highest burden of malaria cases and 93% of all malaria cases reported are from this region. More than half of these cases come from only six countries, namely: Nigeria (25%), Democratic Republic of Congo (12%), Uganda (5%), Cote d’Ivoire, Mozambique, and Niger (4%) [1].

In 2018, Mozambique reported 8,921,081 malaria cases and 1114 fatalities, making it one of the leading causes of morbidity and mortality in the nation compared to other countries [2]. The malaria burden in Mozambique is still unacceptably high, trending in the wrong direction in some areas.

Manica province is in the central region of Mozambique and recorded 821,775 malaria cases in 2019, which is the fourth highest number of cases in the country [3].

Sussundenga District in Manica Province has documented high *P. falciparum* incidence at the local rural health centers (RHC). Sussundenga is a rural district located along the Mozambique, Zimbabwe border with differing control policies on each side of the international border.

For malaria control, indoor residual spraying (IRS), insecticide-treated bed nets (ITNs), and parasitological diagnosis in health facilities using rapid diagnostic test (RDTs) with effective artemisinin combination therapy (ACT) are the malaria interventions currently being used in the country [4].

Mathematical modelling can be used for malaria trend predictions, to describe multiple scenarios, to combine strategies for interventions and to provide a verifiable prediction on what can be expected form implemented schemes [5].

Rural health centers (RHCs) in Mozambique have a large volume of time series case data that feeds the Mozambican management information system (SIS-MA) for systematic dissemination of health data from all districts [6]. These data are retrospective and generally only detect patterns after they have happened. Using this real-time data for mathematical modelling can describe current case trends and predict future malaria cases. Modelling can produce valid results and is inexpensive. This can be helpful for planning malaria control and eradication efforts [7].

There is a growing need for methods for accurate forecasting of malaria cases, and in the past years, methods have been developed and produced worldwide. Most of them use monthly data. Monthly data may not capture detailed events and inappropriate measures may be taken. Malaria is impacted by weather which varies weekly. It is common in Mozambique for the weekly precipitation to fluctuate between 0 mm to 400 mm and have average temperatures below 18 °C and 24 °C. These events will greatly influence the mosquito breeding and malaria infection in the following weeks. Malaria epidemics can occur when climate and other conditions suddenly favor transition [8].

Malaria risk is rarely uniform considering households in a village, villages in a district or districts in a country [9], and the establishment of spatiotemporal patterns in malaria maps is necessary for change contextualization [10]. Geographical information systems can help to describe variations in malaria occurrence and identify areas of high risk, assisting in timely interventions [11]. The Malaria Atlas Project uses global data to address critical malaria questions such as the global malaria landscape, its change, and the impact of malaria interventions [10].

Very few studies on temporal and spatial trends in malaria cases have been reported in Mozambique, especially using weekly surveillance data. Understanding these underlying trends and variations are very important for planning and timing interventions at the local scale for the highest impact.

This study’s objective is to analyze the *P. falciparum* temporal and spatial variation to model its pattern using weekly data throughout Sussundenga District and to predict the expected malaria occurrence for application of timely prevention and control interventions.

## 2. Material and Methods

### 2.1. Study Area

Sussundenga is a district of Mani Province in Western Mozambique with a land area of 7107 square kilometers and a population of approximately 168,000 [12]. It lies between latitudes 19°00′ and 20°30′ South and longitude 32°30′ and 34°00′ East. It borders the districts of Manica and Gondola to the north through to the Revué and Zònue rivers to the south, with the district of Mossurize and the province of Sofala to the east and the district of Buzi (Sofala Province) to the West with the Republic of Zimbabwe [13], as presented in Figure 1.

The inhabitants of the district are mainly rural citizens, with 20% under four years old and less than 3% greater than 65 years old. The majority of the population lives in traditional huts (90%) with less than 1% with piped water, 77% with no latrine access, and less than 3% with electricity in 2011 [12]. Sussundenga is administratively divided into four wards of Sussundenga—Sede, Dombe, Moha and Rotanda. The district has 15 villages and 13 rural health centers (RHCs).

The district has two seasons, the rainy season from November to March and the dry season from May to August. The remaining months represent a transition period between the two seasons. The average annual rainfall increases with altitude, varying from 800 mm in low altitude less than 200 m, up to 1400 mm and more in the in mountainous areas. The altitude varies from 200 to 1500 m. Average summer temperatures (rainy season) are approximately 21 °C throughout the District. Warmer conditions, with temperatures around 25 °C, occur in the Dombe Administrative Post. In Sussundenga in the plateau and mountainous areas in the west of the District, lower temperatures from 15 to 17.5 °C occur [14].

### 2.2. Study Subjects

Public RCHs in Sussundenga collect daily *P. falciparum* malaria case data that are compiled to produce the Weekly Epidemiological Bulletin (BES). This is sent to the planning division of Sussundenga District for the District weekly epidemiologic bulletin. This surveillance system was established in 2015 and is part of the national malaria control program (PNCM). The data was collected from 12/13 RHCs, from 2015 to 2019. Malaria-positive cases were captured using mostly rapid diagnostic tests (RDT) and microscopy for diagnosis. The data reports confirmed cases into two age groups: under five years old and over five years old (Appendix A). The population data used the annual population projection from the National Institute of Statistics of Mozambique [15] from 2015 to 2019. Before data analysis, missing data was calculated by imputation using multivariate normal procedures [16]. A schematic representation of data flow and analysis is presented in Figure 2.

### 2.3. Data Analysis

Analysis of variance (ANOVA) was performed to determine differences between weeks and years using Tukey’s test for mean separation. The model used was:
(1)Yij=μ+τi+βj+γij+ϵijk
where μ is the overall mean response, τ*i* is the effect of month *i*, *βj* is the effect the *j*-th of factor B (week) and Y*ij* is the effect of any interaction between the *i*-th level of A and the *j*-th level of B, and τ*i* is the effect of month *i* [17].

*P. falciparum* incidence per 100 person-years was calculated by dividing the total number of cases by total population and then multiplying by 100 [18].
(2)$$Incidencerate=\frac{Total\;number\;of\;cases}{Total\;population}\;\;\times \;100$$

To obtain groupings or clusters of similar weeks—a hierarchical cluster—a summary of distance matrix analysis was performed. The hierarchical cluster analyses followed three basic steps: (1) calculate the Euclidian distances using a 1.0 cluster cutoff for the weeks, (2) link the clusters and, (3) choose a solution by selecting the right number of clusters [19] resulting in a dendrogram, a tree-based two-dimensional plot [19]. For the preparation of choropleth maps of the malaria incidence in Sussundenga District, data on the administrative division of the country were acquired from the Mozambique National Mapping Center (CENACARTA) [20]. The data from the administrative division were clipped and added to the incidence data of malaria between the age groups using ArcGIS 10.7.1 (ESRI, Redlands, CA, USA) [21].

The ARIMA model has three values, namely “*p*” weeks over P period for each of the 52 weeks sets in our data set, differencing over “*d*” adjacent weeks or D periods, and moving averages sustained over “*q*” weeks or Q periods [7]. The *p* and *q* are the number of significant lags of the autocorrelation function (ACF) and the partial autocorrelation function (PACF) plots, respectively, and *d* is the different order needed to remove the ordinary non-stationarity in the mean of the error terms [21,22].

The ARIMA model using the Box–Jenkins method was performed in three principal steps:

(1) Model identification tentative from the ARIMA class. (2) Estimation of parameters in the identified model. (3) Diagnostic checks.

Model identification tentative: In this step, graphical devices, namely autocorrelation function (ACF) and partial autocorrelation function (PACF), were used as guides to select one or more Autoregressive Integrated Moving Average (ARIMA) models.Estimation of parameters in the identified model: In this step, the precise estimate of the coefficients of the model was selected, chosen from the identification step.Diagnostic check: This step was used to determine if the estimated model was statically adequate. If the identified model passed the diagnostic check, the model was ready to be used for forecasting. If it failed, the model was modified through a new cycle process.

After obtaining a stationary series, a basic model can be identified. There are three basic models, AR (autoregressive), MA (moving average) and their combination, ARMA. When regular differencing is applied in conjunction with AR and MA, they are referred to as ARIMA, with the “I” meaning “integrated” [23].

The model used was:
(3)$$Yt=\theta_{0}+\phi_{1}y{\;}_{t-1}+\ldots +\phi {\;}_{p}\;y{\;}_{t-p}+\varepsilon {\;}_{t}-\theta_{1}\;\varepsilon {\;}_{t-1}-\ldots -\theta {\;}_{q}\;\varepsilon_{\;t-q}$$
where *y_t_* is the real value at the of *t*, *θ* and *ϕ* are the moving average and autoregressive coefficients, respectively, *p* and *q* are integer numbers referencing the order of the autoregressive and moving average, respectively, *ε_t_* is the error, and *d* is the differencing parameter [7].

Data were tested for stability plotting the temporal trend and tested using the Mann-Kendall method. The Bartlett test was used to assess normality, and the Dick-Fullert test was performed [24] to determine stationarity. Seasonal differencing was applied to remove the seasonal trend. Plots of auto correlation function (ACF) and partial auto correlation functions (PACF) were created to determine the trends in the data. The lowest values of Akaike information criteria (AIC) and short Bayesian criteria (SBC) were used to assess the goodness of the fit among identified models. Ordinary least-mean squares (OLS) were used for model estimation. For a diagnosis check, residual plots of ACF, PACF, and Portmanteau test were used [25]. The forecasting was performed for 52 weeks in 2016. The statistical analysis was carried out using SPSS IBM version 20 and NCSS Data analysis 2020.

## 3. Results

### 3.1. Descriptive and ANOVA

Over the 260 weeks (from 2015 to 2019), 372,498 cases of *P. falciparum* were recorded in Sussundenga, 177,957 from children under five years old (47.5%) and 194,541 (52.2%) from individuals over five years old (Figure 3).

The average weekly cases for children under five years old was 680 standard deviation (Sd) = 250.1 and 748 (Sd = 320.1) for individuals over five years old. The highest weekly cases were 1263 cases in children under five years old at week 9 and 1797 cases in individuals over five years old at week 5 in 2017. The lowest weekly malaria cases were recorded in 2018 with 30 and 19 cases for children under five years old and individuals over five years old, respectively, both at week 32.

The year 2019 had the highest number of cases for both age groups, with 41,887 and 50,326 cases in the children under five years old and individuals over five years old, respectively. Analysis of variance (ANOVA) indicated a statistically significant difference in malaria incidence between years and weeks for both categories (*p* < 0.05), Table 1A,B.

### 3.2. Temporal Clusters

Figure 4 presents the weekly malaria temporal clusters for under five and over five years old. For under five years old, three malaria clusters were identified: cluster one of five weeks, from week 31 to 35, where the lowest cases of malaria occurred, on average 394 sd 168.9 per week. The second cluster, from weeks 20 to 47, with a moderate number of cases, on average 570 sd 173.2 cases per week. Cluster three, from week 48 to week 19, with highest malaria cases, on average 855 sd 202.4 weekly.

For individuals over five years old, two temporal clusters were identified: cluster one, from week 48 to 24, with the highest malaria cases, on average 846 sd 272.4 weekly cases and cluster two, from week 23 to 47, with a moderate number of cases, on average with 516 sd 171.4 cases.

### 3.3. Malaria Incidence Spatial Variation for under Five Year Old Category

Figure 5 presents the choropleth maps of spatial distribution of malaria incidence in the last five years and the average cases in children under five years old. The maps show the high concentration of malaria incidence in the Sussundenga-Sede RHC and low concentration of malaria incidence in RHCs in Rotanda and Dombe. The Moha admirative post had a moderate concentration of malaria incidence. Overall, children under five years old in the Dombe and Rotanda administrative posts presented 0.5 episodes of malaria per year, while residents of the Moha administrative post had about one case of malaria per year and residents of Sussundenga-Sede had more than one case of malaria per year.

### 3.4. Malaria Incidence Spatial Variation for 5< Years Category

Figure 6 presents the choropleth maps of spatial distribution of malaria incidence variation from 2015 to 2019 in individuals over five years old. The maps indicate a moderate concentration of malaria incidences in the Sussundenga-Sede, Rotanda and Moha admirative posts and a low concentration of malaria incidence in the admirative posts of Dombe. On average, patients in this category experienced 0.27 cases of malaria per year. Overall, one in six residents in Dombe, Rotanda and Moha administrative posts will have an episodes of malaria per year, while one in three residents of Sussundenga-Sede will experience an episode case of malaria per year.

### 3.5. ARIMA Modelling (under Four Years Old)

Figure 7A presents the trends for the temporal series plot for under four years old. There is a decreased number of malaria cases in ages 0–4 years old, confirmed by the Mann-Kendall test, Sen’s slope = −0.521. The series presents several peaks and fluctuations; the weekly peaks are separated by more than a few weeks, suggesting a seasonal pattern.

The Bartlett test indicated normality for the data, Jack Bera = 4.35, P = 0.114, DF = 2 and no transformation was needed. The Dick-Fuller test indicated a non-stationarity of the mean Tau = 4.35, P = 0.114, DF = 2 and first-order differencing was applied, and a stationary pattern obtained (Figure 7B). Figure 7C,D presents the features of the data for autocorrelation (ACF) and partial autocorrelation (PACF) plot. The ACF indicates an exponential decay and PACF with a single spike at lag 1, suggesting an ARMA (2,1) for non-seasonal and ARMA (1,1) for seasonal components. In eight experiments, the selected final model for malaria patients under five years old was ARIMA (2,2,1) (1,1,1) _52_ and the trend equation was:
(4)Xt =741.0547−0.4336641×(week)

All the coefficients were statically significant at 0.05. Table 1A presents the goodness-of-fit results.

For a diagnostic check, the residual ACF and PACF plot (Figure 7F,G) both indicated that all the terms were within to the confidence intervals, implying that the residuals are “white noise”. The Portmanteau test of the residuals also indicated the model adequacy, Portmanteau = 13.35, P = 0.42, DF = 13.

Using Equation (1), the malaria cases for under four-year-old patients were forecasted for year 2016 (Table 2). Figure 7E presents the predicted trends and intervals (90% and 95%).

Figure 8A presents the trends for the temporal series plot for individuals over five years old. There is an increasing tendency of malaria cases in over five-year-old patients confirmed by the Mann-Kendall test, Sen’s slope = 0.100. The series presents several peaks and fluctuations; the weekly peaks are separated by more than few weeks, suggesting a cyclical pattern.

The Bartlett test indicated normality for the data, Jack Bera = 17.17, P = 0.1011, DF = 2. The Dick-Fuller test indicated a non-stationarity of the mean, Tau = −3.96, P = 0.118, DF = 2, and first-order differencing was applied and a stationary pattern obtained (Figure 8B). Figure 8C,D presents the features of the data for the autocorrelation (ACF) and partial autocorrelation (PACF) plot. The ACF indicates an exponential decay and PACF with a single spike at lag 1.

In eight experiments, the selected final model for malaria patients older than five years old was ARIMA (2,2,1) (1,1,1) _52_ and the trend equation was:(5)$$X_{t}=(715.3047)+(0.2524547)\times \left(\mathrm{week}\right)$$

All the coefficients were statically significant at 0.05. Table 1B presents the goodness-of-fit results for over five years old. For a diagnostic check, the residual ACF and PACF plot (Figure 8F,G) both indicated that all the terms were interior to the confidence limit, implying that the residuals are “white noise.” The Portmanteau test of the residuals also indicated the model adequacy to lag 16, Portmanteau = 13.29, P = 0.4253, DF = 13. Using Equation (1), the malaria cases for 0–4-year-old patients were forecasted for year 2016 (Table 3). Figure 8E presents the predicted trends and intervals (90% and 95%).

## 4. Discussion

Malaria epidemiology and its modelling has had limited investigation in Sussundenga District. The age category under five years old reported 52.5% of the total malaria cases, although this category represents only 21% of the population of the district. This imbalance is attributed to immunity. Most malaria cases and death occur in young children due to low immunity. Partial immunity is developed over years of exposure. Although it does not provide complete protection, it reduces severe malaria in adults [26] and this can explain the higher cases in the under-five category and lower cases in the over-five category.

Malaria occurrence varies by week and year in Sussundenga, and the first weeks of the year (weeks 5 and 9) presented the highest number of cases while week 32 (September) presented the least cases. January to Mach are the months that record higher amounts of rain and humidity in Sussundenga, while September is the month with least humidity [14]. This pattern was also reported in Chimoio, Beira and Maputo in Mozambique [27,28,29]. The disproportional peak of malaria in 2019 occurred when cyclone IDAI hit the area and the annual rainfall recorded in Sussundenga was almost double of annual average [30].

The malaria incidence in this study was 91.3 per 100 for children under five years old and 27.3 per 100 in individuals older than five years old. In Chimoio, an average of 20.5 per 100 persons malaria incidence was reported, in Manica Province it was 43 per 100 persons and 33 per 100 persons in Beira [27,28,29] in Mozambique.

This study identified three malaria clusters of malaria cases for children under five years old. Higher number of cases coincides with the rainy season, a lower number of cases with the dry season, and moderate cases with the transitional period. For individuals over five years old, only two cluster were identified, one with high malaria cases coinciding with the rainy season and another cluster with lower malaria cases in the dry season. A similar pattern was reported in Chimoio and Maputo [31,32] in Mozambique. In Senegal, 23% of malaria positive cases were reported during rainy season and 9% during the dry season [33]. Seasonal differences in infectious diseases were also reported in central Europe [34].

This study found temporal and spatial variation of malaria cases between administrative wards (posts). Rotanda administrative posts present lower malaria episodes per year, while Sussundenga-Sede administrative posts have more episodes of malaria per year. This can be a result of the differences in average annual rainfall, in altitude, and in temperature between the different administrative posts. Rotanda is located at an altitude around 1500 m (highland), and temperatures range from 15 to 17.5 °C. Sussundenga-Sede has an altitude at around 600 to 800 m (midland) with an average temperature of 21 °C, and Dombe has an altitude between around 200 (lowland) with an average temperature of around 25 °C.

Variation in malaria cases as a result of environmental conditions were also reported in Beira, Chimoio, and Maputo in Mozambique [29,31,32].

In the present study, children under five have on average 0.9 episodes of malaria per year and individuals over five years old have 0.27 episodes of malaria per year. This can be related to lower immunity in children under five years old. In Manhiça, Mozambique in 2008, an average of more than two episodes of malaria per year were reported [33]. In Malawi, an overall incidence rate of clinical malaria of 1.2 cases per child per year was reported [34]. Studies in South Africa and Zimbabwe reported opposite results, with children under five having a lower positivity rate. In Zimbabwe, 95% of malaria cases were among individuals older than five years old [35,36].

This study finds a decreased trend in number of malaria cases for under five year olds and an increasing tendency for malaria cases in patients older than five years old. The decreasing trend in this category was also reported by the World Health Organization for Mozambique [3]. In Tanzania, the peak prevalence shifted from children 5–9 years old to those aged 10–19 years old [37]. The effective interventions against malaria lead to age-shifts, delayed morbidity, or rebounds in morbidity and mortality [38]. Pregnant woman and their newborns receive an ITN at the pre-natal clinic. Decreasing cases in this age range and shifts to higher ages with clinical malaria may be an indicator of effective malaria control.

A rapid decline in malaria burden was reported in many areas in Africa from 2015 to 2017, with incidence declining by 27.9% and mortality by 42.5%. Despite the declines, 90.1% of Sub-Saharan African residents live in endemic regions (Cameron).

Similar results were reported in Chimoio Municipality [23], Mozambique in general [39], and the Limpopo region in South Africa [21], contrary to decreasing tendencies in the neighboring countries [36,37,38]. This can be due to different approaches in malaria interventions. In Mozambique, the malaria surveillance is mostly passive case detection in the health clinics while other regions use active case detection in the communities [39].

Both qualitative and quantitative bioinformatics models are advocated for forecasting and public health promotion [40]. In Ampá Brasil, 10 deterministic and stochastic statistical models were tested to predict malaria cases from 1997 to 2016. Deterministic models performed better, and the ARIMA model was the best for predicting future scenarios [41].

An agent-based model validated malaria incidence data collected in Chimoio, Mozambique [42]. In recent years an increase usage of time series techniques has enabled better results [42].

The ARIMA (2,2,1) (1,1,1) _52_ models in this study provide the optimal approach to forecast malaria cases per week over the years for different age categories. The goodness of fit was 68.15% for malaria cases of children under five years old and 73.2% for malaria cases of individuals older than five years old. In Chimoio in 2018, a similar study indicated an ARIMA (2,1,0) (2,1,1) model with a goodness of fit of 72.5% [12]. In the Philippines, ARIMA (2,1,0) was found to be the appropriate model to predict malaria incidence using weekly data [43]. Similar studies were carried out in Ghana, Afghanistan, Nigeria, Zambia, and India using monthly data with comparable results [7,43,44,45,46].

The ARIMA model of this study is robust, inexpensive, and can predict the expected number of malaria cases. This model can contribute to timely prevention and control planning measures such as awareness campaigns, correct times and places to spray, and elimination of vector breeding places. As a result, malaria reduction can occur, saving lives and improving the livelihood of Sussundenga residents. Nowadays, with the advent of GIS, computers, and more real-time data available, weekly planning is advised, reducing, for example, medicine waste and expiring test kits, especially in developing countries where resources for health services are scare.

## 5. Limitations of the Study

This study result may be over- or underestimated since there are generally underreported cases, especially from places lacking health centers, while some cases patients may be diagnosed more than once a year, be self-medicated, or use traditional healers [23]. The weekly malaria missing data was 13%, and missing data proportion is directly related to the quality of statistical inference, although there is no established cutoff regarding an acceptable percentage of missing data for valid statistical inferences [44]. One advantage of using the ARIMA approach is the relative simplicity and stability of the model in predicting malaria cases in a context where political unrest and poor resources lead to a lack of detailed data [7].

## 6. Conclusions

The findings indicate that cases are decreasing among children under five years old and are increasing slightly in individuals older than five years old. The *P. falciparum* case occurrence presents a weekly temporal and spatial pattern, peaking during the wet season. Based on the temporal distribution and modelling using ARIMA, more efficient strategies that target this seasonality can be implemented to reduce the overall malaria burden in the Sussundenga District and regionally. The model can be used to test other infectious diseases, and other models should also be considered.

## Figures and Tables

**Figure 1 ijerph-18-05692-f001:**
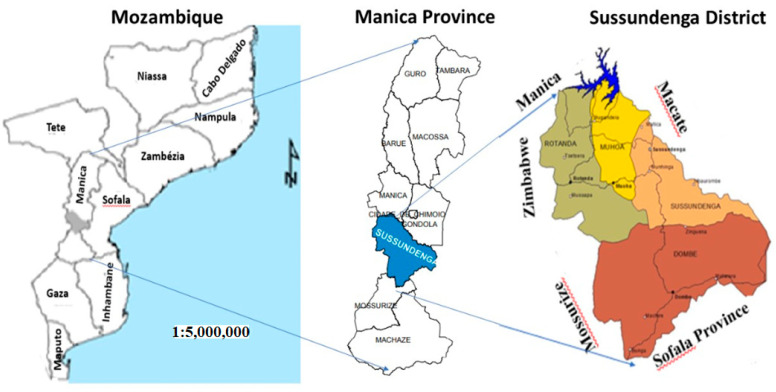
Study area. Adapted from Cenacarta, 2011.

**Figure 2 ijerph-18-05692-f002:**
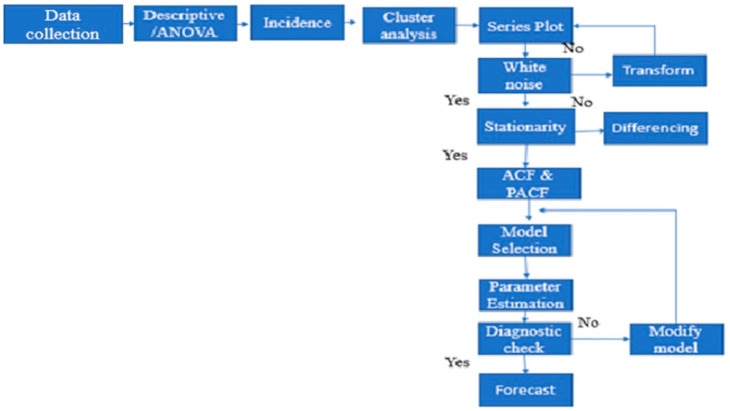
Schematic representation of data flow and analysis.

**Figure 3 ijerph-18-05692-f003:**
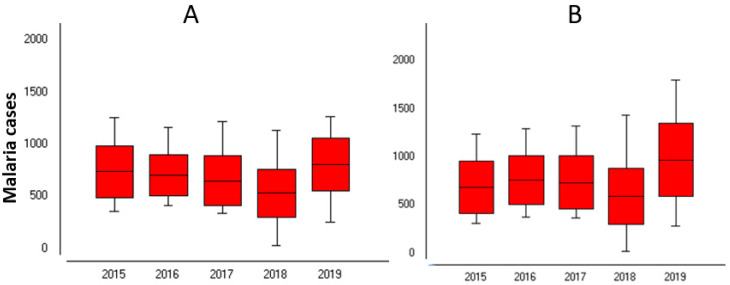
Malaria cases in Sussundenga 2015–2019. (**A**) Under 5 years; (**B**) Over 5 years old.

**Figure 4 ijerph-18-05692-f004:**
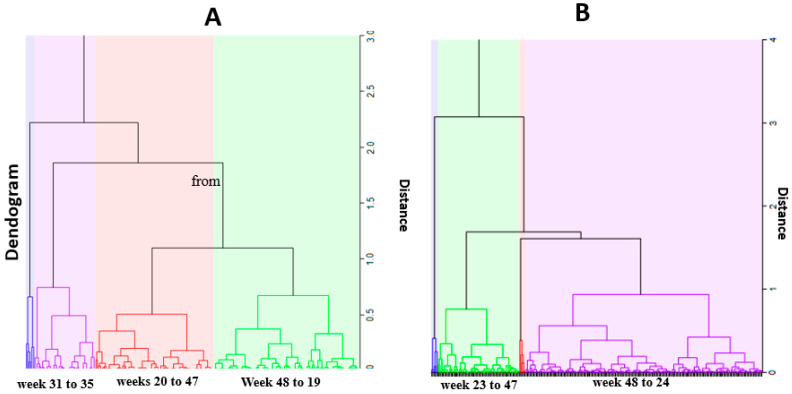
Temporal clusters. (**A**) under 4 years; (**B**) over 5 years.

**Figure 5 ijerph-18-05692-f005:**
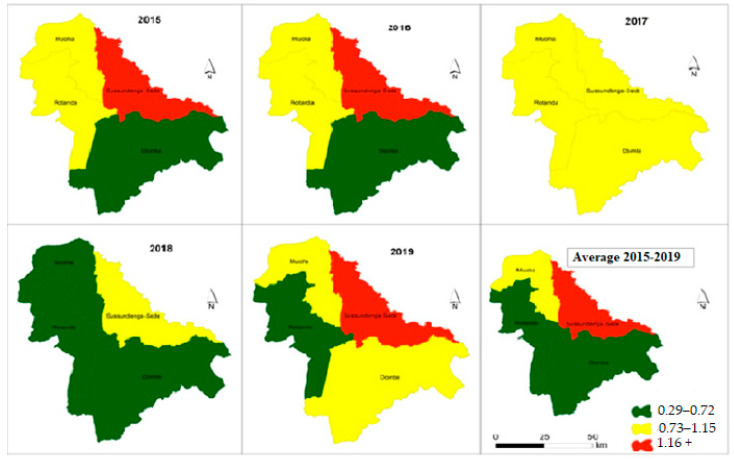
Incidence malaria variation in children under 5 years in Sussundenga 2015–2019.

**Figure 6 ijerph-18-05692-f006:**
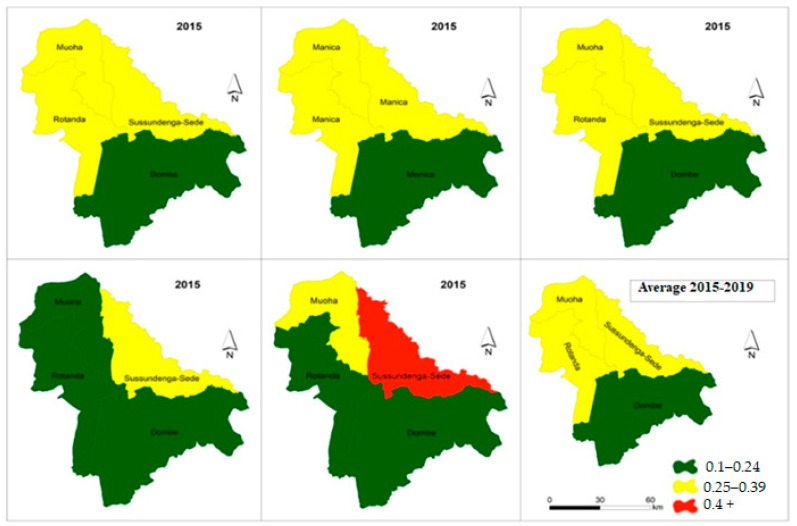
Incidence malaria variation in individuals over 5 years in Sussundenga 2015–2019.

**Figure 7 ijerph-18-05692-f007:**
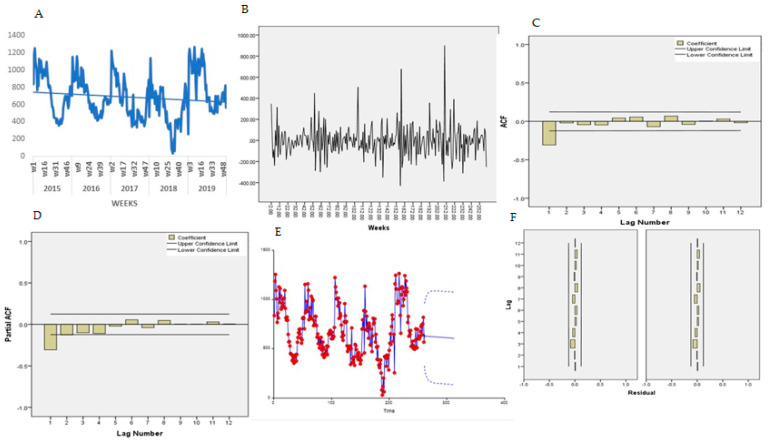
Trend of malaria 0 to 4 years. (**A**) Malaria trend, (**B**) first order differencing, (**C**) ACF, (**D**) PACF, (**E**) predicted trend intervals, (**F**) ACF and PACF residuals.

**Figure 8 ijerph-18-05692-f008:**
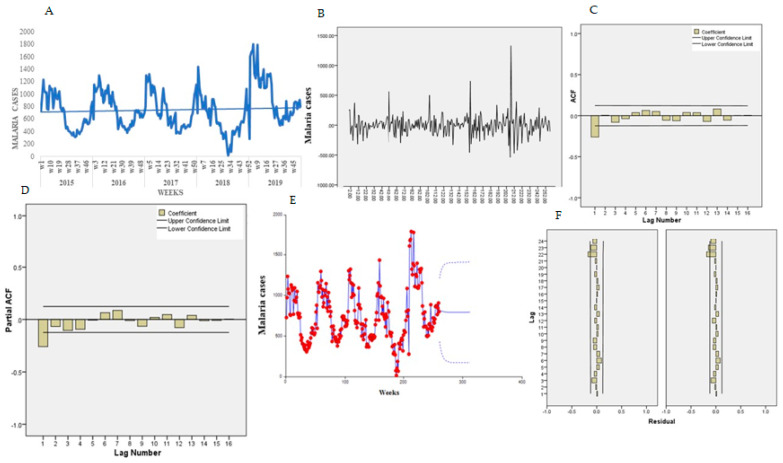
Trend of malaria over 5 years old. (**A**) Malaria trend, (**B**) first order differencing, (**C**) ACF, (**D**) PACF, (**E**) predicted trend intervals, (**F**) ACF and PACF residuals.

**Table 1 ijerph-18-05692-t001:** Analysis of variance (ANOVA) of malaria occurrence between weeks and years in Sussundenga. (A) Under five years old, (B) over five years old.

**A**					
**Year**	**2015**	**2016**	**2017**	**2018**	**2019**
Weeks	52	52	52	52	52
Mean	737 ^a^	700 ^b^	648 ^c^	531 ^c^	806 ^d^
Sd	246	194	236	228	252
**B**					
**Year**	**2015**	**2016**	**2017**	**2018**	**2019**
Weeks	52	52	52	52	52
Mean	687 ^a^	760 ^a^	735 ^a^	592 ^b^	968 ^c^
Sd	270	247	276	289	374

^a,b,c^ different letters indicate difference between years.

**Table 2 ijerph-18-05692-t002:** Under 5 years. Malaria cases forecast for year 2020 in Sussundenga.

Week	Forecast	LL 95%	UL 95%	Week	Forecast	LL 95%	UL 95%
1	337	115	560	27	309	−23	641
2	325	74	576	28	308	−24	641
3	326	56	596	29	308	−24	640
4	324	40	609	30	307	−25	640
5	323	28	619	31	307	−25	639
6	322	19	626	32	307	−26	639
7	321	11	631	33	306	−26	638
8	320	5	635	34	306	−27	638
9	319	1	638	35	305	−27	638
10	319	−3	640	36	305	−28	637
11	318	−6	642	37	304	−28	637
12	317	−9	643	38	304	−28	636
13	316	−11	644	39	304	−29	636
14	316	−13	644	40	303	−29	635
15	315	−14	644	41	303	−30	635
16	314	−15	644	42	302	−30	635
17	314	−17	644	43	302	−31	634
18	313	−18	644	44	301	−31	634
19	313	−18	644	45	301	−31	633
20	312	−19	644	46	301	−32	633
21	312	−20	643	47	300	−32	633
22	311	−21	643	48	300	−33	632
23	311	−21	643	49	299	−33	632
24	310	−22	642	50	299	−33	631
25	310	−22	642	51	299	−34	631
26	309	−23	641	52	298	−34	631

LL = Low limit, UL = Upper limit.

**Table 3 ijerph-18-05692-t003:** Over 5 years Malaria cases forecast for year 2016 in Sussundenga.

Week	Forecast	LL 95%	UL 95%	Week	Forecast	LL 95%	UL 95%
1	821	432	1210	27	790	175	1406
2	812	370	1255	28	790	174	1406
3	811	331	1290	29	790	174	1406
4	808	300	1316	30	790	174	1406
5	806	276	1336	31	790	174	1407
6	804	256	1351	32	790	174	1407
7	802	241	1363	33	791	174	1407
8	800	228	1372	34	791	174	1407
9	799	218	1379	35	791	175	1407
10	797	210	1385	36	791	175	1407
11	796	203	1389	37	791	175	1408
12	795	198	1393	38	791	175	1408
13	794	193	1396	39	792	175	1408
14	794	190	1398	40	792	175	1408
15	793	187	1399	41	792	175	1408
16	792	184	1401	42	792	176	1409
17	792	182	1402	43	792	176	1409
18	792	181	1403	44	793	176	1409
19	791	179	1403	45	793	176	1409
20	791	178	1404	46	793	176	1409
21	791	177	1404	47	793	177	1410
22	791	176	1405	48	793	177	1410
23	790	176	1405	49	794	177	1410
24	790	175	1405	50	794	177	1410
25	790	175	1405	51	794	178	1411
26	790	175	1406	52	794	178	1411

LL = Low limit, UL = Upper limit.

## Data Availability

Data are available as Appendix A.

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
