# Peer review of "Malaria Temporal Variation and Modelling Using Time-Series in Sussundenga District, Mozambique"

_ijerph, 2021, doi:10.3390/ijerph18115692_

Round 1

Reviewer 1 Report

The paper describes the use if an ARIMA model to describe weekly malaria case surveillance data from Sussundenga District in Manica, Mozambique. The objectives are clearly laid out, and the model is justified, though the specifics of the model and its use in malaria could be better described. The background should be strengthened, and the context of the study site should be discussed more thoroughly so that results of the model can be understood better. For example, what are some specific stratification approaches are now apparent that could help more effectively target interventions?

Additional specific comments include:

  • A few formatting and syntax errors to clean up throughout – as an example, at the end of the sentence in line 39 the word ‘each’ might be clearer inside the parentheses: ‘…Côte d’Ivoire, Mozambique, and Niger (4% each)1.’ Also, watch the accent in Côte d’Ivoire. Also, in Line 99 – In Portuguese, the abbreviation for the NMCP in Mozambique is PNCM.
  • I think the descriptions of (1) greater malaria burden among some age groups compared to others and (2) increasing malaria burden over time are confused throughout.

Abstract

  • Line 21: Children under 5 years had increased malaria cases in comparison to 5 years and older, or had a greater incidence of malaria cases?

Background

  • Line 34: The numbers used to make the comparison to COVID-19 incidence are already comically outdated. Also, it looks like this statement may be confusing confirmed malaria cases with estimated malaria cases. Consider removing this comparison.
  • Line 42: Though malaria burden in Mozambique is still unacceptably high, and trending in the wrong direction in some regions, it’s not accurate to say that ‘little progress has been made over the past 20 years’. Look more carefully at the reference provided (including the WMR link), and consider referencing https://malariaatlas.org/trends/country/MOZ, the latest WMR 2020, and Bhatt et al. Nature. 2015 Oct 8;526(7572):207-211. doi: 10.1038/nature15535. Epub 2015 Sep 16, as well.
  • Line 43: Manica province recorded 821,775 malaria cases during which time period – 2018?
  • Line 49: I think the modelling may have estimated the impact of dramatic increases in intervention coverage, and not their feasibility.
  • Line 70: The last sentence in the paragraph is confusing. Monthly data may capture detailed events and inappropriate measures may be taken? Also, while weather does vary weekly (actually, weather varies daily and hourly as well), the biology of malaria transmission – which is dependent on the slightly longer developmental and life cycle traits of mosquitoes and parasites – does not vary on quite these same time scales. Also, given variations in health care utilization and reporting rates, it might be argued that monthly aggregates and monthly trends could be a more accurate – and actionable – approach. An expanded discussion of the utility of a weekly approach would be interesting here.
  • Some discussion of surveillance DQA, Dx and Tx commodity stock outs and availability, and how these things may have changed over time – is needed. How might this influence the model and its interpretation? Also, what malaria control interventions were in place during the study period, and were they consistent throughout?

Methods

  • Line 87 – this report is from 2011. If there aren’t any more recent SES estimates, maybe acknowledge that this data was collected more than 10 years ago.
  • Line 94 – does this data include cases diagnosed by local APEs?

Discussion

  • The discussion could be more focused and directed. For example, in the first paragraph the discussion starts by describing the concentration of cases in the 0–4-year-old category by percentage of cases vs. proportion of population in Sussundenga but shifts abruptly to discuss the case incidence rate in How are these things related?
  • The age categories could be better defined throughout as well – is this 0-59 months? 0- 5 years in some cases and 0-4 years in others?
  • Line 316 – How would decreased transmission from effective control lead to an increasing tendency for malaria cases in patients older than five years old?

Author Response

Reviewer 1

Dear Reviewer. Thanks for the comments and they are very useful for the article imrovemente. Thanks also for the Malaria Atlas Project, references. Indeed I also have articles in Maparia Risk Maping and Modelling and they will be useful for further studies.

Comments and Suggestions for Authors

  1. The paper describes the use if an ARIMA model to describe weekly malaria case surveillance data from Sussundenga District in Manica, Mozambique.
  2. The objectives are clearly laid out, and the model is justified, though the specifics of the model and its use in malaria could be better described.

Response: Thanks for the observation. Better description of model was decribed.

  1. The background should be strengthened, and the context of the study site should be discussed more thoroughly so that results of the model can be understood better.

Response: Thank you very much for the observation. A major revision in the background was carried out.

Additional specific comments include:

  1. A few formatting and syntax errors to clean up throughout – as an example, at the end of the sentence in line 39 the word ‘each’ might be clearer inside the parentheses: ‘…Côte d’Ivoire, Mozambique, and Niger (4% each)1.’ Also, watch the accent in Côte d’Ivoire. Also, in Line 99 – In Portuguese, the abbreviation for the NMCP in Mozambique is PNCM. Corrected

Response: Thank you very much. This was corrected.

  1. I think the descriptions of (1) greater malaria burden among some age groups compared to others and (2) increasing malaria burden over time are confused throughout. T

Responde: We do agree, thanks very much. Correction was made.

  1. Abstract
    • Line 21: Children under 5 years had increased malaria cases in comparison to 5 years and older, or had a greater incidence of malaria cases?

Response: Thanks for the observation, correction was made.

  1. Background
    • Line 34: The numbers used to make the comparison to COVID-19 incidence are already comically outdated. Also, it looks like this statement may be confusing confirmed malaria cases with estimated malaria cases. Consider removing this comparison.

Response: Thans for the observation. This was removed

  • Line 42: Though malaria burden in Mozambique is still unacceptably high, and trending in the wrong direction in some regions, it’s not accurate to say that ‘little progress has been made over the past 20 years’. Look more carefully at the reference provided (including the WMR link), and consider referencing https://malariaatlas.org/trends/country/MOZ, the latest WMR 2020, and Bhatt et al. Nature. 2015 Oct 8;526(7572):207-211. doi: 10.1038/nature15535. Epub 2015 Sep 16, as well.

Response: Thanks for the observation. This was corrected

  • Line 43: Manica province recorded 821,775 malaria cases during which time period – 2018?

Response: Agree with th observation and the year was added

  • Line 49: I think the modelling may have estimated the impact of dramatic increases in intervention coverage, and not their feasibility.

Response: Agree and was corrected.

  • Line 70: The last sentence in the paragraph is confusing. Monthly data may capture detailed events and inappropriate measures may be taken?

Response: Agree with observation and this was removed

  • Also, while weather does vary weekly (actually, weather varies daily and hourly as well), the biology of malaria transmission – which is dependent on the slightly longer developmental and life cycle traits of mosquitoes and parasites – does not vary on quite these same time scales.

Response: Thanks for the observation. One very dry or very wet week can result in tremendous increase or decreases in malaria cases.  Extreme events such as IDAI cyclone proved that. With the advent of new information systems and real time data, weekly forecating can be more appropriate. We added that in the discussion.

  • Also, given variations in health care utilization and reporting rates, it might be argued that monthly aggregates and monthly trends could be a more accurate – and actionable – approach.

Response: Agree but, with more real time data for example SISMA) weekly planning can be possible. This will be included in the discussion.

  • An expanded discussion of the utility of a weekly approach would be interesting here.

Response: Thanks for the observation, this was included in the discussion.

  • Some discussion of surveillance DQA, Dx and Tx commodity stock outs and availability, and how these things may have changed over time – is needed.

Response: Thanks for the observation. I am not sure if I understand clearly the point but we included SISMA

  • Also, what malaria control interventions were in place during the study period, and were they consistent throughout?

Response: Thanks for the observation. This was included.

  1. Methods
    • Line 87 – this report is from 2011. If there aren’t any more recent SES estimates, maybe acknowledge that this data was collected more than 10 years ago.

Response: Thanks for the observation. We could not find more recent SES for Sussundenga and we accnwolde it in the article.

  • Line 94 – does this data include cases diagnosed by local APEs?

Response: Yes.  

  1. Discussion
    • The discussion could be more focused and directed. For example, in the first paragraph the discussion starts by describing the concentration of cases in the 0–4-year-old category by percentage of cases vs. proportion of population in Sussundenga but shifts abruptly to discuss the case incidence rate in How are these things related?

Response: Thanks for this very useful observation. We reogarnise the discussion

  • The age categories could be better defined throughout as well – is this 0-59 months? 0- 5 years in some cases and 0-4 years in others?

Response: We do agrre with the observation. We corrected the age groups and are now uniform.

  • Line 316 – How would decreased transmission from effective control lead to an increasing tendency for malaria cases in patients older than five years old?

Response: Thanks for this good question. Indeed, a shift in age category due to malaria control is well documented and, we included in our discussion.

Reviewer 2 Report

This study is of strong practical significance and can serve for malaria control. The analysis of this paper is relatively interesting, and it is suggested that the author should make a more complex analysis in combination with the geographical elements, such as weather in further research. From the current state of this paper, the innovation is slightly insufficient. It is suggested that the author increase the comparison of different prediction methods and analyze the results. In addition, if the author can distinguish the characteristics of age group or the occupation, and then make a detailed analysis, the reference value of the article can be further improved. To sum up, the author is suggested to deepen the depth of analysis or model comparison according to the suggestions. The reviewer's opinion is that this article should be revised and reviewed again.

Author Response

Reviewer 2

Dear Reviewer. Thanks very much for the comments and they are very useful for the article imrovement.

Comments and Suggestions for Authors

  1. This study is of strong practical significance and can serve for malaria control.

Response: Thank you very much for the observation.

  1. The analysis of this paper is relatively interesting, and it is suggested that the author should make a more complex analysis in combination with the geographical elements, such as weather in further research.

Response: Rsponse: Thank you very much for the observation and this suggestion will be taken in account in future research.

  1. From the current state of this paper, the innovation is slightly insufficient. It is suggested that the author increase the comparison of different prediction methods and analyze the results.

Response: Thanks for the observation. We do think that mathematics can contribute for the success of malaria control programs specially in developing countries. One advantage of using ARIMA approach is the relative simplicity and stability of the model in predicting malaria cases in a context where political unrest and poor resources lead to a lack of detailed data. We included this in the article.

  1. if the author can distinguish the characteristics of age group or the occupation, and then make a detailed analysis, the reference value of the article can be further improved.

Response: Thnaks for the observation. We used historical data and they do not provide characteristics of age group or the occupation. For future studies with interviesws the suggestion will be taken in consideration.

  1. To sum up, the author is suggested to deepen the depth of analysis or model comparison according to the suggestions.

Response: Thanks for the observation. A major revision was carried out in line with the observations.

Round 2

Reviewer 2 Report

Although the comments of the first review were not accepted, this article is generally interesting and can be published.